# Popular Religion, Sacred Natural Sites, and "Marian Verdant Advocations" in Spain

**Jaime Tatay-Nieto** [1,*] **and Jaime Muñoz-Igualada** [2]

[1] Facultad de Teología, Universidad Pontificia Comillas, Madrid 28015, Spain
[2] Tragsatec, Asistencia Técnica de la Subdirección General de Biodiversidad y Medio Natural, Ministerio para la Transición Ecológica, Madrid 28071, Spain; at_sgmn2@mapama.es
[*] Correspondence: jtatay@comillas.edu; Tel.: +34-915-422-800

**Abstract:** A relevant number of shrines, hermitages, monasteries, and pilgrimage routes in Spain are located within or near Natura 2000, a European network of protected core breeding and resting sites for rare and threatened species, and some rare natural habitat types. Given the growing interest in alternative conservation strategies and the geographical correlation between nature preserves and Sacred Natural Sites (SNS), this paper explores how religious devotions have made preservation possible in Spain. By an extensive literature review and interviews with long-established custodians of nonurban Marian sanctuaries, it looks at the development of plant-related allegorical titles, the multiple meanings of "Marian verdant advocations", and the role popular religion has played in connecting theological insights with particular elements of natural ecosystems helping value and preserve the Spanish biocultural heritage. We found that 420 Marian titles directly refer to plant species or vegetation types and many of the nonurban Marian sacred sites are placed in well-preserved natural areas, some of them playing a human-related added value for most emblematic National Parks, like the sanctuaries of El Rocío (Doñana NP) and Covadonga (Picos de Europa NP). We conclude that there is a strong relationship between popular religion, Marian verdant titles, and nature conservation.

**Keywords:** Plants; popular religion; biocultural diversity; Natura 2000; Mariology; Christianity; religion and nature

## 1. Introduction

Around the world, protected areas have been established on the sites of existing or former sacred natural sites (Verschuuren et al. 2008). Most of these sites preceded the establishment of protected areas, often by many centuries, sometimes by millennia (Mallarach et al. 2014). Scientists, managers, and conservation practitioners have recognized that the sacredness of nature underpins the world's first conservation areas, which often were Sacred Natural Sites (SNS), or sacred landscapes (Frascaroli 2013; Verschuuren and Brown 2018). There is evidence across the world that territories managed by monastic communities over the centuries have been more carefully preserved than the surrounding ones (Mallarach et al. 2016) and ancient religious themes and practices underpin contemporary environmental discourses and initiatives (Adler 2006; Berry 2015). Moreover, sacred plants and animals, through their spiritual meanings, have played a role in the development of a sacred geography and in the maintenance of "biocultural diversity" (Posey 1999; Bhagwat and Rutte 2006; Pungetti et al. 2012), interpreted as "the diversity of life on earth in both nature and culture" (Pungetti 2013).

The values, religious beliefs, and management practices of traditional peoples are increasingly considered important elements of effective community-based natural resource management (Cox et al. 2014; Ostrom 1990). Many SNS form "a largely unrecognized shadow conservation network"

(Dudley et al. 2008). There is evidence that sacredness can be a powerful means of conservation when linked to customary institutions and a broadly respected belief system (Virtanen 2002) and interreligious advocacy campaigns have been instrumental in the global efforts to preserve ecosystems, species, and a stable climate (Oviedo 2012; Tatay and Devitt 2017). Partnerships between religious and conservation groups represent "significant untapped potential" (McLeod and Palmer 2015), which can promote and sustain conservation efforts. As some scholars have argued, despite the historical divide, there is a growing "convergence of beliefs" (Dudley and Higgins-Zogib 2012) between faith groups and conservation practitioners. However, in order to better articulate the interests of both communities, the latter has to understand and address the needs and aspirations of faith groups, whereas the former should recognize conservation priorities and rethink theologically their duties to the natural world.

In Spain, there are approximately 12,300 churches, shrines, sanctuaries, and pilgrimage sites (Batalla Gardella 2002): 1200 (10%) are named after a Christological title, 4300 (35%) after a Marian devotion, and 6800 (55%) are related to many different saints. A significant share of the nonurban religious sites is located in natural preserves of high ecological value and has played a prominent role in the "sacred spatial planning" (Muñoz and Miguel 1997) of the Iberian landscape. Given the growing interest in customary institutions and types of protection that differ from those promoted via legal mechanisms (Colding and Folke 2001; Colding et al. 2003; Bhagwat and Rutte 2006; Swiderska 2011), the significant number of Christian sacred sites and pilgrimage routes placed within or near Natura 2000 areas, and the deep popular piety associated with Mary, we focused our exploration on the religious and cultural practices underpinning Marian titles and pilgrimages to SNS in Spain as a preliminary mapping of the terrain, a way of providing examples of effectively managed community areas that have preserved valuable ecosystems, traditions, and beliefs for centuries. This study aims at breaking ground on a topic that had not been previously explored and expects to open new research possibilities.

It is necessary to specify that by "Marian verdant title" we mean a noncanonical, poetic or allegorical way of depicting Mary of Nazareth in popular religion using an arborescent or plant-related name. In an analogous manner, by "Marian verdant advocation" (from the Latin *advocatio*, pleading the cause of another) we mean a complementary botanical term that applies to Mary referring to a certain mystery, virtue, or attribute of her, to special moments in her life, to places linked to her presence or to the discovery of an image of her. Nuestra Señora del Espino (Our Lady of the Hawthorn), Virgen de la Encina (Virgin of the Holm Oak), Mare de Déu del Roser (Mother of God of the Rose), Nuestra Señora del Olivo (Our Lady of the Olive), and Virgen del Olmo (Virgin of the Helm) are some of the most popular Marian verdant advocations in Spain.

## 2. Marian Popular Devotions and The Religious Significance of Plants

According to Robert Harrison, within the Western cultural imagination trees and forests "remain the correlate of human transcendence." They also evoke, however, a fear of "the disappearance of boundaries, without which the human abode loses its grounding" (Harrison 1992, p. 247). This dual, ambiguous, and paradoxical meaning of forests—the complex experience of *tremens* and *fascinans* conveyed by the natural world—also permeates popular religion in Spain and is reflected in the rituals and devotional practices associated with rural Marian verdant advocations. In fact, most of these advocations trace their origin to a place in the forest or in a cave where a shepherd, a monk, or a child either found a hidden image of Mary or had a personal, fortuitous, transcendent encounter—an apparition of the Mother of God.

As Ronald David Lawler has noted, "the very nature of Catholic faith inclines the believer to consider private revelation a real possibility" (Lawler 1984, p. 103). Since the first visions of Mary in the 4th century, there have been an estimated 21,000 Marian apparitions across the world (Warner 2016), and despite secularization, their frequency has not diminished in the 20th century. Indeed, most Marian verdant advocations have emerged from popular devotions associated with mysterious private revelations. "Mary's decision to appear at particular places at particular times is one of the things that

most clearly establishes her as a distinct supernatural personality" (Carroll 1986, p. 224). The figure of Mary mediates these numinous encounters in the wilderness between a humble, devout person and the divine. In these intimate, personal encounters, the transcendent–immanent line, as well as the human–nature divide, is often blurred. Many interpretations have been offered in the last decades, from the Freudian understanding of apparitions as hallucinations intended to gratify childhood experiences or traumas to a missing feminine aspect in Christian theology to a sign of a spiritual void (Horsfall 2000; Carroll 1986).

Despite secularization, Mary still has a status in Western society and Marian devotions play a prominent role in popular religion (Romero 1993; Warner 2016). Marian titles and advocations are also widely used. The role and significance of Mary in popular Catholicism, however, is a matter of intense academic debate. Warner 2016 has described five prototypical Marian images—virgin, queen, bride, intercessor, and mother—whereas Roten (1994) identified six types of Marian popular imagery—naturalist, poetic, exotic, essentialist, abstract, and miraculous. According to Roten, miraculous images are "the oldest and most widespread form of popular Marian imagery. This type of image is closely linked to apparitions of the past, to famous sanctuaries and pilgrimages" (Roten 1994, p. 108). This is certainly the case with most rural Marian advocations in Spain (Batalla Gardella 2002). The mother image, the predominant type of Marian verdant title, mediates the presence of the Creator and acts as a type of theophany, an immanent presence of a life-giving force. Popular religion acknowledges a communal participatory role in the understanding and interpretation of the central beliefs of a particular faith. As Hilda Graef puts it: "In popular devotions, the believer is called to participate rather than be instructed" (Graef 2009, p. 427). In Latin Christianity, the symbol of Mary as a life-giving mother, intercessor, and mediatrix (between a creator God or mother Goddess and humankind) plays a prominent role in many popular devotions (Romero 1993). However, "Mary is not herself the source of life, but she leads or points to that source of life" (Roten 1994, p. 112). Through these devotions, believers are transformed in the way they see, interact, and value the natural world. The many Spanish Marian arborescent titles are a reflection of the way the transformation works.

The tree of life metaphor, a component of the world tree motif, was common in Assyrian and Egyptian imagery and is also present in the Hebrew Bible (Gen. 2:9; Prov. 3:18; 11:30; 13:12; 15:4) and the New Testament (Rev. 2:7; 22:2). Adam and Eve are not only placed in a garden, they are set in a dialogical relationship with one tree that qualifies in two ways: the tree of life and the tree of knowledge of good and evil. Both symbolic trees have a distinct function, the tree of knowledge as a test, and the tree of life as a reward for obedience (Mettinger 2007).

The woman of the Apocalypse, later interpreted by the Christian church as Mary, the new Eve, does not appear in a paradisiacal Garden. However, at the very end, the final eschatological vision draws us back to the very beginning, the Genesis tree of life motif (Rev. 22:2). Popular religion may have found inspiration in this ancient religious motif often setting apparitions in a forest or in a garden, channeling the life-giving force of trees through the use of arborescent titles. It is also relevant to note the symbolic function of trees as an indestructible, living force and the evidence that, except for the pine trees, all Marian arborescent titles in Spain (including *Pinus canariensis*, a species that resists fire) refer to species capable of new growth, sprouting from the branches or the base of burnt or cut trunks. This may also explain the relative absence of conifers, most of which are not fire-prone.

The Modern Catalan advocation to the Mare de Déu del Roser (Mother of God of the Rose) deserves a special mention not only for its popularity, but also because of its theological meaning (Wilkins 1969). As Sara Horsfall explains: "The visionaries encounter Mary, a being who converses with them. Many of the visionaries, and those who witness the visions, experience the environment as well. A frequent report is the smell of flowers of roses" (Horsfall 2000, p. 379). The garden rose bush is the most common Marian verdant title in Spain, deriving its name from the Latin *rosarium* (literally 'rose garden'), which, "in fourteenth-century Europe had also come to signify a collection of devotional texts ('roses') offered in praise" (Mitchell 2009, p. 5). At the end of the Middle Ages, "numerous allegorical works began to appear with titles like 'Little Rose Garden,' 'Spiritual Flower

Garden,' or 'Rose Garden of the Heart.' When the garden has become the soul—or the soul a rose garden—the image of picking spiritual flowers from it to offer to the Virgin is a logical step. From there, it is only a small step to the concept of the rosary" (Winston-Allen 2005, p. 99). Significantly, whereas the advocation to the Virgen del Rosario (Virgin of the Rosary) is widespread across Spain, only the Catalan-speaking regions have kept its floral resonance.

Thorny bushes, very common in the semi-arid Mediterranean biome, carry a strong Biblical resonance and have been interpreted in many cultures as symbols of spiritual transformation. When it comes to one of the central Marian advocations, the *mater dolorosa* (sorrowful mother), it is inevitable to draw a theological parallel between the crucifixion scene and the tradition of the crown of thorns. We do not know for sure from which species of bush or tree the Roman soldiers tore twigs to elaborate the crown of thorns (cf. Mark 15:17; Matthew 27:29; John 19:2). Botanists and historians have identified a number of shrubs as potential candidates (Evans 2014, p. 138–39): Palestine buckthorn (*Rhamnus palaestinus*), Jerusalem thorn (*Paliurus spina-Christi*), boxthorn (*Lycium* spp.), common thorny burnet (*Sarcopoterium spinosum*), Christ-thorn (*Ziziphus spina-Christi*). Three of these species—Jerusalem thorn, boxthorn, and Christ-thorn—thrive in the Iberian Peninsula, though they are not very common. Similar shrubs and trees, however, are widespread. The reason why so many (37.4%) Marian apparitions in Spain have taken place on top, inside, or near a small thorny shrub (*Rosa* spp., *Crataegus monogyna*, *Rubus ulmifolius*, *Erica* spp.) or tree with prickly foliage (*Quercus ilex*) remains unclear, but their strong Biblical, Christological, and cultural associations seem a plausible explanation.

Another ancient Christian tradition, the eremitic-monastic, has related the seven virtues of cloistered life with the seven trees mentioned by Isaiah (Is. 41:18–9): fir or pine, cedar, hawthorn, myrtle, olive, box, and elm (Pungetti et al. 2012). Interestingly, five out of the seven—*Pinus*, *Crataegus*, *Olea*, *Buxus*, *Ulmus*—are well represented among the Spanish Marian verdant advocations. Furthermore, there is a significant overlap between the 20 sacred trees identified in the Celtic Ogham alphabet (Graves 1876) and the verdant advocations analyzed here. The Celtic culture deeply shaped the Iberian Northwest before the arrival of the Romans and may have played a role in translating the ancient meaning of certain trees (*Malus*, *Crataegus*, *Quercus*, *Ilex*, *Vitis*, *Populus*, *Taxus*) into Christian symbols via popular religion.

As Judith Crews has pointed out, "tree worship has for the most part disappeared from the modern world. However, the symbols that remain in language, lore and culture serve as reminders of the rich relationship between human thought and the forest world" (Crews 2003, p. 43). Across the world, different forms of tree symbolism "embody different ways of relating to place and conservation practices" (von Hellermann 2016). This seems to be the case in Spain too, where monasteries and rural sanctuaries have played a key role in articulating a "sacred landscape" (Cinquepalmi and Pungetti 2012) or "sacred spatial planning" (Muñoz Jiménez 2010). Moreover, "in the location of the Spanish rural sanctuaries, some clear geographical clichés converge (the mountain, the sickle, the river, the fountain, the island, the peninsula, the singular tree, the hanging stones, the forms of wind origin, etc.), as well as several historical circumstances such as primitive hermitages, castles of military orders, uninhabited areas, ancient ruins, dolmens and megaliths, cemeteries, place of birth of a saint, etc." (Muñoz and Miguel 1997, p. 109).

It is no coincidence that a significant number of highly popular Marian sanctuaries in Spain are located inside National Parks (NP) and other types of nature preserves within Natura 2000 (Table 1). The sanctuaries of El Rocío (Doñana NP), Covadonga (Picos de Europa NP), Montserrat (Parc Natural de Montserrat) and Peña de Francia (Parque Natural de las Batuecas-Sierra de Francia) are some paradigmatic cases of Marian shrines placed in ecologically very valuable settings. In Southern Europe, as in many other regions of the world, "yesterday's sacred grove is today a biosphere reserve, a natural heritage site or protected area" (Crews 2003, p. 43).

**Table 1.** Popular Spanish Marian sanctuaries with long-established custodians within Natura 2000.

| Christian Sacred Site. | Nature Preserve (Natura 2000) | Province |
|---|---|---|
| Ermita de la Mare de Déu dels Lliris | Parc Natural de la Font Roja | Alicante |
| Santuario de la Virgen de Covadonga | Parque Nacional de Picos de Europa | Asturias |
| Monestir de Montserrat | Parc Natural de la Muntanya de Montserrat | Barcelona |
| Monasterio de Santo Toribio de Liébana | Parque Nacional de Picos de Europa | Cantabria |
| Ermita de la Virgen de la Cueva Santa | Parque Natual de la Sierra Calderona | Castellón |
| Ermita de Nuestra Señora de los Reyes | Parque Natural de la Dehesa | El Hierro |
| Santuari de la Mare de Déu de Núria | Parc Natural de les Capçaleres del Ter i el Freser | Girona |
| Santuario de la Virgen de la Salud | Parque Natural del Barranco del Río Dulce | Guadalajara |
| Santuario de la Virgen de Arantzazu | Espacio Natural de Aizkorri-Aratz | Guipúzcoa |
| Ermita de la Virgend de Lourdes | Parque Nacional de Garajonay | La Gomera |
| Ermita de la Virgen del Pino | Parque Nacional de la Caldera de Taburiente | La Palma |
| Monasterio de Nuestra Señora de Valvanera | Parque Natural de la Sierra Cebollera | La Rioja |
| Ermita de Nuestra Señora de los Dolores | Parque Nacional de Timanfaya | Lanzarote |
| Ermita de la Virgen de Navalazarza | Espacio Natural de la Dehesa de Moncalvillo | Madrid |
| Santuari de la Mare de Déu de Lluc | Paratge Natural de la Serra de Tramuntana | Mallorca |
| Monasterio de Leyre | Sierra de Leyre | Navarra |
| Santuario de la Peña de Francia | Parque Natural de Las Batuecas-Sierra de Francia | Salamanca |
| Monestir de Poblet | Parc Natural de les Muntanyes de Prades i Poblet | Tarragona |
| Santuario de la Virgen de las Nieves | Parque Nacional del Teide | Tenerife |
| Santuario de la Virgen del Tremedal | Espacio Natural de los Montes Universales | Teruel |

## 3. Research Methods

The approach of this study is threefold and complementary. First, we systematically searched bibliographical references to Catholic sacred sites related to Marian pilgrimages, shrines, and sanctuaries in Spain. We were able to identify a total of 17 publications, mostly consisting of travel guidebooks with valuable anthropological, ethnographic, and theological comments (Abad León 1990; Amengual i Batle 1997; Caballero Mújica 1999; Carrasco Terriza 1992; Carreres i Péra 1988; Cebrián Franco 1989; Delclaux 1991; Fernández Álvarez 1990; Fernández Sánchez 1994; Fernández-Ladreda 1989; Ferri Chulio 2000; González Echegaray 1993; Iturrate 2000; Llamas 1992; López Martín 1998; Sánchez Ferrer 1995; Torra de Arana 1996).

Second, these bibliographical sources and the conversations with long-stablished custodians—priests, religious communities or lay fraternities—of nonurban Marian sanctuaries (Table 1) provided a total of 420 locations, which we inventoried and geo-located with the objective of assessing both the spiritual values and the cultural practices associated to a particular site. Maps were generated using ARCGIS tools.

Third, frequency distribution and descriptive statistics of the different verdant advocations were calculated from the database of sacred sites compiled at the beginning of the study. The answers of the interviews and the results of the bibliographical research and the statistical analyses were discussed, so that a better comprehension of the interaction between popular piety, verdant advocations, theological meanings, and geographical distribution could be achieved. The religious significance and theological underpinnings of the different verdant advocations were investigated. The final results were synthesized and presented to a peer group and debated among its members. Conclusions were extracted from these dialogues.

## 4. Results

As one of the targets of the study was to roughly explore the potential correlation between the geographical distribution of green Marian titles and the EU Natura 2000, we first investigated the location of the 420 sacred sites and, then, their corresponding adscription to different municipalities of Spain. After grouping and correcting the misleading toponyms and the imprecise references to

equivocal locations, we identified 374 municipalities with one or more sanctuaries in their territory. Of these 374 municipalities, a significant 62.30% (233) are in Natura 2000. By means of ARCGIS tools, it was calculated the precise extension of each municipality included in Natura 2000. The 233 municipalities summed a total of 821,795 ha inside Natura 2000, 30.15% of the total surface of the municipalities compared to 25% for the overall Spanish territory reported by Pozo Rivera et al. (2013). A geographical representation is provided in Figure 1, where the distribution of the municipalities with Marian verdant advocations is shown along with the Spanish Natura 2000 network.

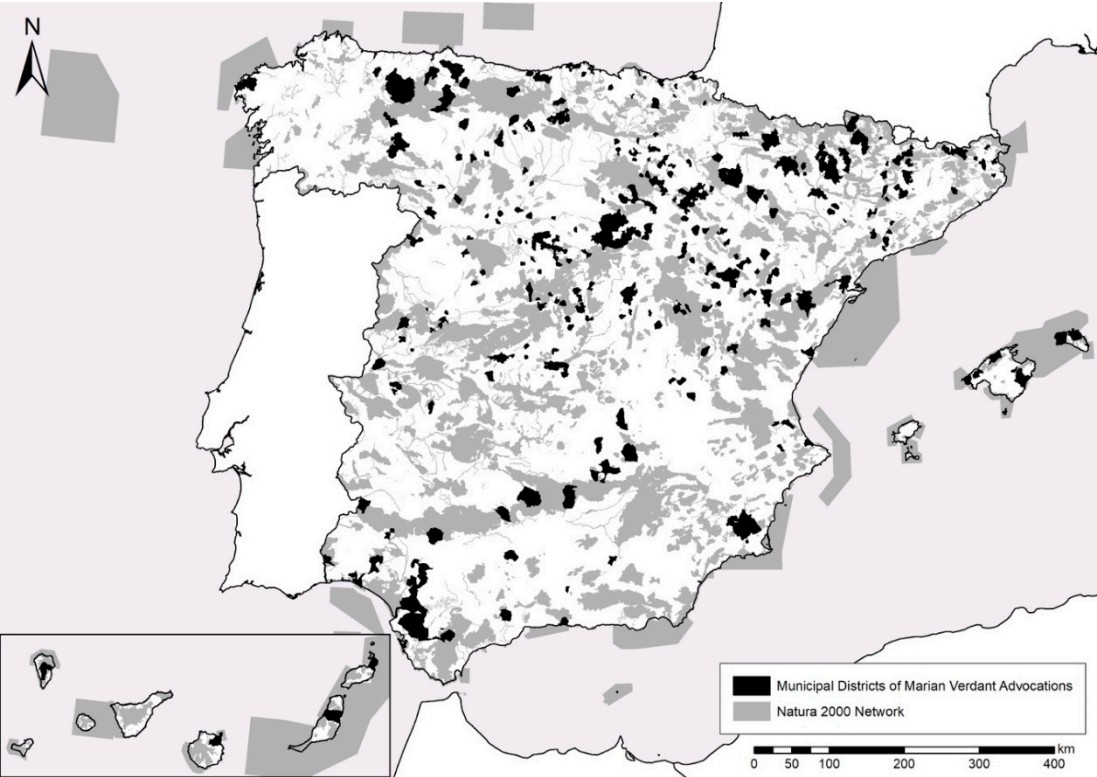

**Figure 1.** Municipal districts with Marian verdant advocations in Spain.

The vernacular names, frequency, and geographical distribution of Marian verdant advocations are, however, very uneven (Table 2; Table 3). Out of the 420, an estimated 88.6% (372) refer to 50 species in 30 families, whereas the remaining 11.4% of the titles (48) are named after a generic plant or vegetation-type: meadow, flower, forest, peat, tree.

**Table 2.** Marian verdant advocations in Spain (n ≥ 20).

| Family. | Species | | n. | Σ |
|---|---|---|---|---|
| Rosaceae | *Rosa* spp. | Rose | 64 | |
| | *Crataegus monogyna* | Common hawthorn | 30 | |
| | *Rubus ulmifolius* | Elmleaf blackberry | 18 | |
| | *Malus domestica* | Apple | 8 | 126 |
| | *Pyrus* spp. | Pear | 3 | |
| | *Prunus avium* | Wild cherry | 1 | |
| | *Prunus dulcis* | Almond | 1 | |
| | *Prunus* spp. | Plum | 1 | |
| Fagaceae | *Quercus ilex* | Holly oak | 38 | |
| | *Quercus* spp. | Oak | 14 | 59 |
| | *Castanea sativa* | Sweet chestnut | 4 | |
| | *Fagus sylvatica* | Beech | 3 | |

**Table 2.** *Cont.*

| Family. | Species | | n. | Σ |
|---|---|---|---|---|
| *Ulmaceae* | *Ulmus minor* | Field elm | 24 | 29 |
| | *Celtis australis* | European nettle tree | 5 | |
| *Oleaceae* | *Olea europaea* | Olive | 22 | 26 |
| | *Fraxinus excelsior* | European ash | 4 | |
| *Pinaceae* | *Pinus* spp. | Pine | 9 | 22 |
| | *Pinus canariensis* | Canary Island pine | 7 | |
| | *Pinus pinaster* | Maritime pine | 4 | |
| | *Pinus halepensis* | Aleppo pine | 1 | |
| | *Pinus sylvestris* | Scots pine | 1 | |
| *Vitaceae* | *Vitis vinifera* | Common grape vine | 20 | 20 |

## 4.1. Frequency of Marian Verdant Advocations

Roses (*Rosa* spp.) and common hawthorns (*Crataegus monogyna*), both thorny shrubs in the family *Rosaceae*, are the first and third most popular Marian verdant advocations in Spain (Table 2), making up almost a quarter of all (22.4%). The advocation to the Mare de Déu del Roser (Mother of God of the Rose) is the single most popular verdant title. These prickly shrubs carry a strong Biblical resonance (Evans 2014, p. 22) and have been interpreted in other Mediterranean countries as symbols of repentance and conversion (Pungetti et al. 2012). Elmleaf blackberry (*Rubus ulmifolius*), also a thorny bush in the family *Rosaceae*, is well represented (4.3%) and may well have conveyed similar meanings and associations in popular piety (Nuestra Señora de la Zarza, Navalazarza, Zarzaquemada, Zarzuela). According to an ancient tradition, Jewish and Christian, both *Crataegus* and *Rubus* have long been identified as possible candidates of the burning bush (Evans 2014, p. 23). The apple tree (*Malus domestica*)—with a clear relation with Eve, the figure upon which Mary, the New Eve (Ryder 1974), is presented in the Gospels—along with *Pyrus* and several *Prunus* represent a significant 3.3% of the total.

The second most common Marian verdant advocation (12.4%) is represented by a large group of evergreens and marcescent oaks (*Quercus* spp.) in the *Fagaceae* family, a highly diversified group of trees. Holly oak or holm oak (*Quercus ilex*), a tree with prickly foliage native to the Western Mediterranean, is the second most frequent Marian verdant title (9%). The many vernacular names referring to *Quercus ilex* (Encina, Encinar, Encinillas, Carrasco, Carrascal, Lluc) reflect the high linguistic and cultural diversity of the country, the many uses of the holly oak and the wide distribution of a species present in every region of the Iberian Peninsula and the Balearic Islands (Rivas Martínez 1987). Marian apparitions in Spain usually take place within a sacred grove (as in Lluc, a toponym derived from the Latin word *lucus*, holly oak grove) or on top of a tree. The European or sweet chestnut (*Castanea sativa*) and beech (*Fagus sylvatica*) are much less frequent and their advocations appear only in the Atlantic North (Virgen del Hayedo, Haya, Faya; Virgin of the Beech) or in particularly humid enclaves within the Spanish Mediterranean bioregion (Nuestra Señora del Castañar; Our Lady of the Chestnut).

Four species represent, alongside *Rubus ulmifolius*, the second most frequent group of green advocations (Table 4): the field elm (*Ulmus minor*), a hardwood whose natural range is predominantly south European (Virgen del Olmo, Olmos, Olmacedo, Olma; Mare de Déu dels Oms, Omedes); the olive tree (*Olea europaea*), a small tree found across the Mediterranean (Virgen de la Oliva, Olivar, Olivares); several species of pine trees (*Pinus* spp.); and the common grape vine (*Vitis vinifera*), the only liana among all advocations (Virgen de las Viñas, Viña, Vid, Viñedo, Parral, Parrales, Mare de Déu del Vinyet). Except for the elmleaf blackberry, all four species in this cluster are or have been (in the case of the field elm, whose populations were devastated by the Dutch elm disease) commercially, culturally, or religiously very significant.

**Table 3.** Marian verdant advocations in Spain (n < 20).

| Family | Species | | n. | Σ |
|---|---|---|---|---|
| *Salicaceae* | *Salix* spp. | Willow | 8 | 12 |
| | *Populus nigra* | Black poplar | 4 | |
| *Poaceae* | *Stipa tenacissima* | Esparto grass | 5 | |
| | Hay | Hay | 4 | 10 |
| | *Phragmites communis* | Common reed | 1 | |
| *Lamiaceae* | *Rosmarinus officinalis* | Rosemary | 9 | 9 |
| *Ericaceae* | *Erica* spp. | Heath | 6 | 7 |
| | *Arbutus canariensis* | Canary madrone | 1 | |
| *Juncaceae* | *Juncus* spp. | Reed | 7 | 7 |
| *Moraceae* | *Morus alba* | White mulberry | 4 | 6 |
| | *Ficus carica* | Fig | 2 | |
| *Araliaceae* | *Hedera helix* | Common ivy | 5 | 5 |
| *Punicaceae* | *Punica granatum* | Pomegranate | 5 | 5 |
| *Aquifoliaceae* | *Ilex aquifolium* | Holly | 3 | 3 |
| *Betulaceae* | *Corylus avellana* | Hazel | 3 | 3 |
| *Cistaceae* | *Cistus* spp. | Rockrose | 3 | 3 |
| *Cupressaceae* | *Juniperus oxycedrus* | Prickly juniper | 3 | 3 |
| *Taxaceae* | *Taxus baccata* | Yew | 3 | 3 |
| *Buxaceae* | *Buxus sempervirens* | European box | 2 | 2 |
| *Fabaceae* | *Ulex parviflorus* | Gorse | 1 | 2 |
| | *Genista* spp. | Broom | 1 | |
| *Liliaceae* | *Lilium* spp. | Lily | 2 | 2 |
| *Apiaceae* | *Foeniculum vulgare* | Fennel | 1 | 1 |
| *Brassicaceae* | *Raphanus sativus* | Radish | 1 | 1 |
| *Chenopodiaceae* | *Beta vulgaris* | Beet | 1 | 1 |
| *Juglandaceae* | *Juglans regia* | Walnut | 1 | 1 |
| *Myrtaceae* | *Myrtus communis* | Mirtle | 1 | 1 |
| *Palmaceae* | *Phoenix dactylifera* | Date palm | 1 | 1 |
| *Polypodiopsida* | *Polypodiopsida* | Fern | 1 | 1 |
| *Rutaceae* | *Citrus aurantium* | Orange tree | 1 | 1 |

A third group of less-frequent advocations (Table 4, Cluster III) refers to species such as rosemary (*Rosmarinus officinalis*), willow (*Salix* spp.), heath (*Erica* spp.), reed (*Juncus* spp.), esparto grass (*Stipa tenacissima*), common ivy (*Hedera helix*) and pomegranate (*Punica granatum*). Surprisingly, rosemary, a woody, perennial herb with fragrant, needle-like leaves in the *Lamiaceae* family, is the only representative in its family among all verdant advocations (Nuestra Señora del Romeral, Romero). This fact requires a special comment, because "in the Iberian Peninsula and Balearic Islands live 290 species in the *Lamiaceae* family, corresponding to 36 genera" (Morales 2000, p. 31). In stark contrast with the wide evolutionary radiation of the family and its cultural and economic relevance, species such as thyme (*Thymus* spp.), mint (*Mentha* spp.) or sage (*Salvia* spp.) have not been considered as plausible advocations. Within this cluster, pomegranate, "one of the seven species of promise (Deut. 8:8) [ . . . ] a metaphor of beauty and desire in the Song of Solomon (Song 4:3)" (Evans 2014, p. 172), represents a regional advocation present only in Southern Spain.

**Table 4.** Most common Marian verdant advocations (≥1%).

| | Species | | Marian Advocation | n. | % |
|---|---|---|---|---|---|
| **Cluster I** | *Rosa* spp. | Rose | Roser, Rosa | 64 | 15.2 |
| | *Quercus ilex* | Holly oak | Encina, Encinar, Encinillas, Carrasco, Carrascal, Lluc | 38 | 9 |
| | *Crataegus monogyna* | Common hawthorn | Espino, Arantzazu | 30 | 7.1 |
| **Cluster II** | *Ulmus minor* | Field elm | Olmo, Olmeda, Olmacedo, Oms, Omedes | 24 | 5.7 |
| | *Olea europaea* | Olive | Oliva, Olivar, Olivares | 22 | 5.2 |
| | *Pinus* spp. | Pine | Pino, Pinar, Pinarejo | 22 | 5.2 |
| | *Vitis vinifera* | Common grape vine | Vid, Viña, Viñas, Viñedo, Parral, Parrales, Vinyet, Raïmat | 20 | 4.8 |
| | *Rubus ulmifolius* | Elmleaf blackberry | Zarza, Zarzuela, Zarzaquemada, Navalazarza | 18 | 4.3 |
| **Cluster III** | *Rosmarinus officinalis* | Rosemary | Romero, Romeral | 9 | 2.1 |
| | *Salix* spp. | Willow | Salcedo, Salceda, Salcedón, Sargar, Saz, Salz | 8 | 1.9 |
| | *Juncus* spp. | Reed | Juncal, Junquera, Junqueres, Xunqueira | 7 | 1.7 |
| | *Erica* spp. | Heath | Brezo, Brezales, Bruguers | 6 | 1.4 |
| | *Stipa tenacissima* | Esparto grass | Atocha, Atotxa | 5 | 1.2 |
| | *Hedera helix* | Common ivy | Hiedar, Yedra, Heura | 5 | 1.2 |
| | *Punica granatum* | Pomegranate | Granada, Granado | 5 | 1.2 |

Finally, there is a large group of rare (n. ≤ 4), local advocations named after different species in 20 families (Table 3). These titles usually derive from a toponym (i.e., Mare de Déu de la Murta, *Myrtus communis*; Nuestra Señora del Rabanal, *Raphanus sativus*) or a very particular historical event (i.e., Virgen de la Granada, *Punica granatum*; Mare de Déu del Lliris, *Lilium* spp.). Eleven very rare advocations appear just once: common reed, gorse, broom, fennel, radish, beet, walnut, myrtle, date palm, fern, and orange tree. It is noteworthy the total absence or scarce number of advocations referring to otherwise well-represented genders in the flora of the Iberian Peninsula and the Balearic and Canary archipelagoes such as poplars (*Populus* spp.), ash trees (*Fraxinus* spp.), laurels (*Laurus* spp.), junipers (*Juniperus*) and some common *Prunus* in the *Rosaceae* family (i.e., plums, cherries, peaches, apricots, almonds).

Significantly, families that play a key role in the Mediterranean landscape (i.e., *Lamiaceae*, *Fabaceae*, *Cistaceae*, *Rhamnaceae*, *Pistaceae*), are culturally relevant (*Palmaceae*, *Cupressaceae*, *Lauraceae*, *Moraceae*) or have a significant economic importance (*Poaceae*, *Salicaceae*, *Rutaceae*) seem to be underrepresented among the Marian verdant advocations. The reasons why popular religion has selected some species and ignored others remain unclear and demand further research.

### 4.2. Vegetation Types

According to Christen C. Raunkiaer's life-form scheme (Raunkiaer 1934), which depends on the place of the plant's growth-point (bud) during seasons with adverse conditions (cold and dry periods), there are four different vegetation types within the Spanish Marian verdant advocations: phanerophytes (86.9%), hemicryptophytes (9%), cryptophytes (2.9%), and therophytes (2.4%).

Despite the diversity of vegetation types present in Spain, most verdant advocations are arborescent titles that refer to trees or shrubs of different heights (phanerophytes), which coincides with a growing

body of evidence pointing toward the symbolic importance of trees, forests, and sacred groves in many cultures across the world (Harrison 1992; Crews 2003; Bhagwat and Rutte 2006; Hooke 2012).

*4.3. Geographical Distribution*

The majority of the Marian verdant advocations in Spain are found in the northern half of the Iberian Peninsula, being much less frequent in the South, Galicia, and the Canary Islands (Figure 2).

Three of the most common species (*Rosa* spp., *Crataegus monogyna*, *Rubus ulmifolius*) are thorny shrubs in the *Rosaceae* family. Although the rose appears also in Andalusia, its distribution is clearly northeastern, centered on Catalonia (Figure 3). Hawthorn and blackberry, however, are more evenly distributed throughout the territory, without concentrating on a particular region. Pine trees (*Pinus* spp.), on the other hand, show a distribution centered on three regions: The Canary Islands, Western Andalusia, and Castile.

The other four most common advocations (*Olea europaea*, *Vitis vinifera*, *Quercus ilex*, *Ulmus minor*) are more evenly spread over the territory (Figure 4). However, while the elm advocation is mostly found in the northern half of the peninsula and the vine in the wine regions of the Duero and Ebro valleys, the holm oak and the olive tree are present in many more regions, including the island of Mallorca.

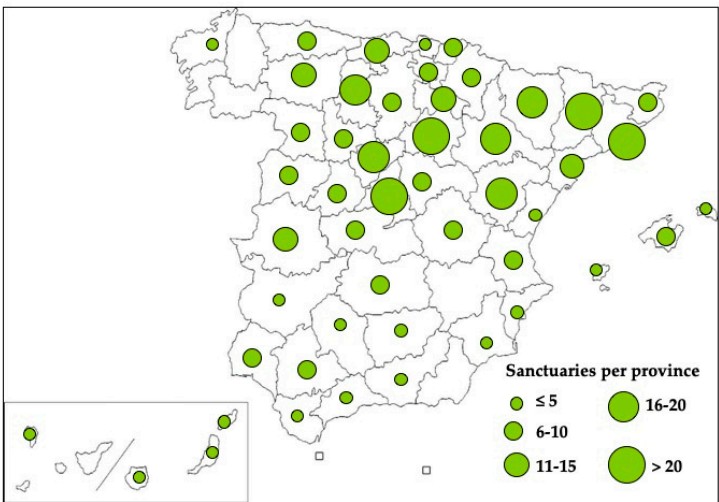

**Figure 2.** Geographical distribution of Marian verdant advocations in Spain.

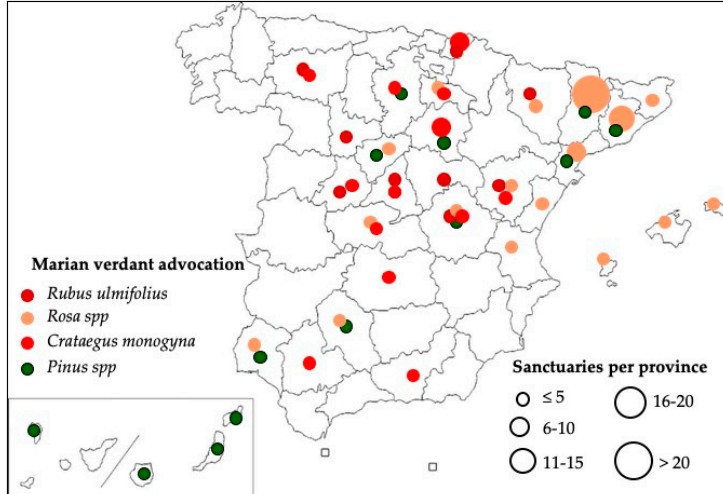

**Figure 3.** Geographical distribution of *Rubus ulmifolius*, *Rosa* spp., *Crataegus monogyna*, *Pinus* spp.

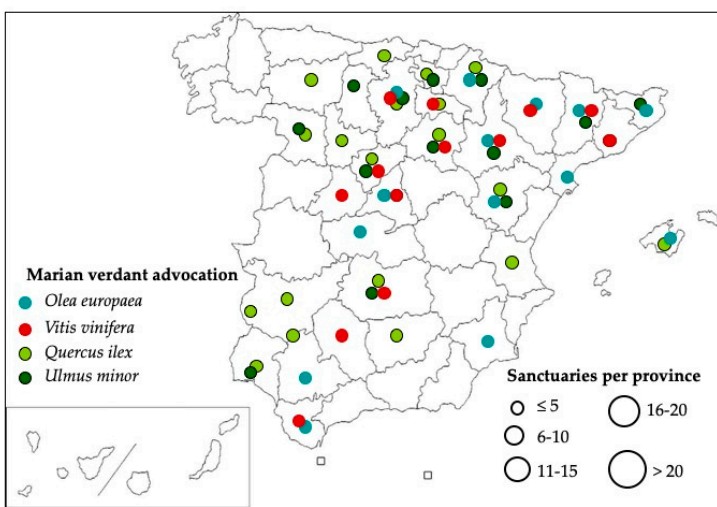

**Figure 4.** Geographical distribution of *Olea europaea*, *Vitis vinifera*, *Quercus ilex*, *Ulmus minor*.

It is not easy to understand why Marian verdant titles are nonexistent or very rare in the southeast and northwest of the Iberian Peninsula. A plausible explanation might be the Church's need to reaffirm the Christian faith by using explicitly theological titles in traditionally Muslim regions (the southeast) or in those areas where the nature-centered cult typical of Celtic spirituality could have generated confusion (the northwest).

## 5. Conclusions

The role of mainstream faiths in preserving SNS and the spiritual values of natural places has been acknowledged by academics (McLeod and Palmer 2015), The World Bank (Palmer and Finlay 2003), environmental think tanks (Gardner 2002), and international conservation agencies (WWF-ARC 2005). This is a relatively recent development that was not expected a few decades ago. For the most part, natural resource management and conservation strategies had been based on secular approaches rooted in the natural sciences (Oviedo 2012). The secular tide, however, seems to be receding and a growing awareness of the links between conservation and spirituality, the recognition of the importance of partnership among all societal actors, and the evidence of complex social–ecological systems (Ostrom 2009; Folke et al. 2016) has led to the inclusion of religious actors in the sustainability debate (Wolf 2017).

Rights-based policies and managerial approaches to preserve valuable ecosystems are not enough. In the international arena, "there has been a renewed interest in the role religions and Faith-Based Organizations (FBOs) can play in fostering sustainability" (Tatay and Devitt 2017, p.123). The growing interest in the role of SNS and popular religion in the context of environmental conservation can be seen as part of a wider effort striving to conserve biocultural diversity (Pungetti et al. 2012) and as a recognition of the importance to commit all societal actors. Existing local Marian verdant advocations are an excellent case study to analyze how traditional religious beliefs can underpin new conservation strategies and preserve biocultural diversity.

Engagement in sustainable ways of living is a key issue to the success of conservation and the survival of civilization as a whole. In this study, we have only focused on the analysis of Marian verdant advocations and other highly popular Marian sanctuaries within or near the Spanish Natura 2000. The limited scope of our research, however, may be a valuable starting point for further studies.

Despite the increasing secularization, the rapid depopulation of rural Spain over the second half of the 20th century, and the disconnect between urbanites and their surrounding landscape, there is a growing interest in spirituality witnessed in recent phenomena such as the revival of ancient pilgrimages (Pack 2010) or the increased participation in traditional devotional practices. There is also a renewed interest in the sacred dimension of the nonhuman world that may help meet contemporary spiritual needs while contributing to articulate ethical arguments for nature conservation. Delving into

the symbolic and sacramental realms can help to explain the connections between traditional value systems and contemporary practices. Lessons learned from the history of SNS may assist in improving the human–nature relationship.

Josep-Maria Mallarach et al. have argued that "the analysis of the criteria applied for the creation and maintenance of conserved areas by Christian monastic communities in diverse ecosystems throughout history is of interest for nature conservation and landscape management" (Mallarach et al. 2016, p. 74). We affirm that this is not only true for monastic communities. A relationship of continuing dialogue between the custodians of religious sites and the managers of protected natural areas can clarify policy in the interest of promoting both the spiritual and cultural values associated with the landscape, and its conservation. If conservation needs to be grounded "in deeply held spiritual, cultural, and aesthetic values and ideas that will engage and inspire people to care for nature over the long term" (Bernbaum 2012, p. 83), then it will be helpful to elicit the deep-seated meanings that sacred sites and religious symbols convey.

A recognition of the existence of many Marian verdant titles by which people have thought and interacted with nature for centuries in Spain has implications for conservation. Understanding how nature-based religious devotions, rituals, and symbols have shaped a particular sacred landscape can inform policies that build on existing local traditions, practices, knowledge, and institutions. As this paper has shown, identifying Marian verdant advocations not only helps to better understand the rich biocultural diversity of Spain, it can also situate these traditions in a particular socioenvironmental history.

**Author Contributions:** conceptualization, methodology, investigation, and writing—review and editing, J.T.-N.; data curation and visualization, J.M.-I.

**Funding:** This research received no external funding.

**Acknowledgments:** The authors thank John Braverman, Roberto Matellanes Ferreras, Stefan Einsiedel, and Michelle Mope Andersson for providing constructive comments which improved the article.

**Conflicts of Interest:** The authors declare no conflict of interest.

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
