# Peer review of "Popular Religion, Sacred Natural Sites, and “Marian Verdant Advocations” in Spain"

_religions, doi:10.3390/rel10010046_

Round 1

Reviewer 1 Report

This is a valuable study that will make an important contribution to research on sacred natural sites. However, the authors can improve the overall clarity throughout, especially for an interdisciplinary audience. 

In general, the authors need to connect the dots on the argument and evidence. What are Marian advocations and verdant titles--this must be explained for an interdisciplinary audience. What is the significance of the various lists of plants? How are advocations and these lists of specific plants related? This needs to be explained much more directly. 

As well, is there evidence of greater biodiversity around these areas? To make the claim that these areas are in better shape, ecologically speaking, due to their sacred connections, evidence of greater-than-normal biodiversity would need to be included. 

Overall, the paper seems out of order. Examples of the religious significance of plants come after the litany of specific species, and the methods are described at the very end. I believe it will help clarity immensely if this is rearranged. Explain the religious significance of these specific plants first then describe research methods, then results of the surveys. Add to this a few sentences/paragraphs describing the meaning and significance of Marian advocations more generally, and I believe this paper will be in good shape and ready to be accepted by the journal. 

I also recommend that the authors consult the work of Fabrizio Frascaroli on Catholic sacred natural sites in Italy. 

Author Response

This is a valuable study that will make an important contribution to research on sacred natural sites. However, the authors can improve the overall clarity throughout, especially for an interdisciplinary audience. 

1.    In general, the authors need to connect the dots on the argument and evidence. What are Marian advocations and verdant titles--this must be explained for an interdisciplinary audience. What is the significance of the various lists of plants? How are advocations and these lists of specific plants related? This needs to be explained much more directly.

We have specified the meaning of Marian advocations/titles in the introduction, defining its meanings and trying to make it clear to a multidisciplinary audience: 

“It is necessary to specify that by “Marian verdant title” we mean a non-canonical, poetic or allegorical way of depicting Mary of Nazareth in popular religion using an arborescent or plant-related name. In an analogous manner, by “Marian verdant advocation” (from the Latin advocatio, pleading the cause of another) we mean a complementary botanical term that applies to Mary referring to a certain mystery, virtue or attribute of her, to special moments in her life, to places linked to her presence or to the discovery of an image of her. Nuestra Señora del Espino (Our Lady of the Hawthorn), Virgen de la Encina (Virgin of the Holm Oak), Mare de Déu del Roser (Mother of God of the Rose), Nuestra Señora del Olivo (Our Lady of the Olive), and Virgen del Olmo (Virgin of the Helm) are some of the most popular Marian verdant advocations in Spain”. 

The theological and cultural significance of the different plants is explained in section 2, right after the introduction. We hope the new articulation helps the reader.

2.    As well, is there evidence of greater biodiversity around these areas? To make the claim that these areas are in better shape, ecologically speaking, due to their sacred connections, evidence of greater-than-normal biodiversity would need to be included. 

We have added a map (Figure 1) and statistical evidence as a way of providing evidence of greater ecological value around Sacred Natural Sites in Spain (4. Results): 

“As one of the targets of the study was to roughly explore the potential correlation between the geographical distribution of green Marian titles and the EU Natura 2000, we first investigated the location of the 420 sacred sites and, then, their corresponding adscription to different municipalities of Spain. After grouping and co            rrecting the misleading toponyms and the imprecise references to equivocal locations, we identified 374 municipalities with one or more sanctuaries in their territory. Of these 374 municipalities, a significant 62.30 % (233) are in Natura 2000. By means of ARCGIS tools it was calculated the precise extension of each municipality included in Natura 2000. The 233 municipalities summed a total of 821,795 ha inside Natura 2000, a 30.15 % of the total surface of the municipalities compared to 25 % for the overall Spanish territory reported by Rivera et al. (2013). A geographical representation is provided in Figure 1, where the distribution of the municipalities with Marian verdant advocations is shown along with the Spanish Natura 2000 network.”

Figure 1.Municipal districts with Marian verdant advocations in Spain

3.      Overall, the paper seems out of order. Examples of the religious significance of plants come after the litany of specific species, and the methods are described at the very end. I believe it will help clarity immensely if this is rearranged. Explain the religious significance of these specific plants first then describe research methods, then results of the surveys. Add to this a few sentences/paragraphs describing the meaning and significance of Marian advocations more generally, and I believe this paper will be in good shape and ready to be accepted by the journal. 

The order has been substantially rearranged, following your suggestion.

4.    I also recommend that the authors consult the work of Fabrizio Frascaroli on Catholic sacred natural sites in Italy. 

We did check out Frascaroli work, it was very helpful indeed.